# Liquid-crystal organization of liver tissue

**Hernán Morales-Navarrete[1†‡], Hidenori Nonaka[1†‡], André Scholich[2†‡], Fabián Segovia-Miranda[1†‡], Walter de Back[3,4], Kirstin Meyer[1], Roman L Bogorad[5], Victor Koteliansky[6,7], Lutz Brusch[4], Yannis Kalaidzidis[1]\*, Frank Jülicher[2,8]\*, Benjamin M Friedrich[8,9]\*, Marino Zerial[1,8]\***

[1]Max Planck Institute of Molecular Cell Biology and Genetics, Dresden, Germany; [2]Max Planck Institute for the Physics of Complex Systems, Dresden, Germany; [3]Institute for Medical Informatics and Biometry, Faculty of Medicine Carl Gustav Carus, Technische Universität Dresden, Dresden, Germany; [4]Centre for Information Services and High Performance Computing, Technische Universität Dresden, Dresden, Germany; [5]David H. Koch Institute for Integrative Cancer Research, Massachusetts Institute of Technology, Cambridge, United States; [6]Skolkovo Institute of Science and Technology, Skolkovo, Russia; [7]Department of Chemistry, MV Lomonosov Moscow State University, Moscow, Russia; [8]Cluster of Excellence Physics of Life, TU Dresden, Dresden, Germany; [9]Center for Advancing Electronics Dresden, Technische Universität Dresden, Dresden, Germany

**\*For correspondence:**
kalaidzi@mpi-cbg.de (YK);
julicher@pks.mpg.de (FJ);
benjamin.m.friedrich@tu-dresden.
de (BMF);
zerial@mpi-cbg.de (MZ)

[†]These authors contributed equally to this work
[‡]The first four authors are listed in alphabetical order

**Abstract** Functional tissue architecture originates by self-assembly of distinct cell types, following tissue-specific rules of cell-cell interactions. In the liver, a structural model of the lobule was pioneered by Elias in 1949. This model, however, is in contrast with the apparent random 3D arrangement of hepatocytes. Since then, no significant progress has been made to derive the organizing principles of liver tissue. To solve this outstanding problem, we computationally reconstructed 3D tissue geometry from microscopy images of mouse liver tissue and analyzed it applying soft-condensed-matter-physics concepts. Surprisingly, analysis of the spatial organization of cell polarity revealed that hepatocytes are not randomly oriented but follow a long-range liquid-crystal order. This does not depend exclusively on hepatocytes receiving instructive signals by endothelial cells, since silencing Integrin-β1 disrupted both liquid-crystal order and organization of the sinusoidal network. Our results suggest that bi-directional communication between hepatocytes and sinusoids underlies the self-organization of liver tissue.
DOI: https://doi.org/10.7554/eLife.44860.001

## Introduction

The liver is the largest metabolic organ of the human body and vital for blood detoxification and metabolism. Its functions depend on a complex tissue architecture. In the lobule, the functional unit of the liver, blood flows through the hepatic sinusoids from the portal vein (PV) and hepatic artery toward the central vein (CV). The hepatocytes take up and metabolize substances transported by the blood and secrete the bile through their apical surface into the bile canaliculi (BC) network, where it flows toward the bile duct near the PV (*Boyer, 2013*; *Meyer et al., 2017*; *Treyer and Müsch, 2013*). Hepatocytes are polarized cells with a unique organization of apical and basal plasma membrane on their surface (*Müsch, 2014*; *Treyer and Müsch, 2013*). In contrast to simple epithelia, where the cells have a single apical surface facing the lumen of organs, hepatocytes exhibit a multi-polar organization, that is, they have multiple apical and basal domains (*Gissen and Arias, 2015*; *Müsch, 2014*; *Treyer and Müsch, 2013*). Such organization allows the hepatocytes to have numerous contacts with the sinusoidal and BC networks to maximize exchange of substances.

Although the general organization of the liver into distinct millimeter-sized lobules is quite clear, the micro-anatomy of a single lobule is much less understood. Sinusoidal endothelial cells and hepatocytes form a heterogeneous 3D packing of cells and labyrinths of sinusoids and BC without apparent order (*Gissen and Arias, 2015*; *Treyer and Müsch, 2013*). However, the function of sinusoidal and BC networks prompt precise design requirements: each hepatocyte must be in contact with both networks, yet the networks must never intersect. This defines a problem of self-organization to satisfy these competing design requirements. Hans Elias in 1949 pioneered the structural analysis of the mammalian liver tissue (*Elias, 1949a*; *Elias, 1949b*; *Elias, 1949c*; *Elias and Bengelsdorf, 1952*). He proposed a structural model whereby the sinusoids are separated from one another by walls of hepatocytes (one-cell-thick), forming a '*continuous system of anastomosing plates, much like the walls separating the rooms within a building*' (*Elias, 1949b*; *Elias, 1949c*). In his idealized model, Elias proposed that the tissue structure is based on hepatic plates built of alternate layers of polyhedral (decahedra and dodecahedra) cells forming a network of BC and traversed by the sinusoids. The model has been a milestone in the field. However, the analysis underlying the structural model was hampered by the difficulties of reconstructing the 3D tissue structure, which at that time relied on stereological analysis of 2D images (*Elias, 1949b*; *Elias, 1949c*; *Elias, 1971*; *Elias and Bengelsdorf, 1952*). Consequently, the limitations in throughput of 3D reconstructions were a major bottleneck for inferring the rules of structure governing liver tissue. Almost 70 years later, we took advantage of developments in tissue clearing, high-performance microscopy imaging, computer-aided image analysis, and 3D tissue reconstruction to revisit the organizational principles of liver tissue.

## Results

### 3D segmentation and quantitative analysis of liver architecture

To understand liver architecture, from the lobule down to the sub-cellular level, we built a 3D geometrical digital representation of adult mouse liver from confocal images (*Figure 1*) using a multiresolution approach (*Morales-Navarrete et al., 2015*). Mouse livers were fixed, sectioned into 100 μm serial slices, and immunostained for cell nuclei (DAPI), cell borders (Phalloidin), hepatocyte apical plasma membrane (CD13), and the extracellular matrix (ECM, fibronectin and laminin) to visualize the basal plasma membrane of hepatocytes facing the sinusoidal endothelial cells (*Figure 1—figure supplement 1*). Full slices and selected regions-of-interest comprising a whole liver lobule were imaged at low resolution (*Figure 1A–C*) to determine the position of CV and PV as landmarks, which allowed locating and re-imaging individual CV-PV-axes at high-resolution (*Figure 1D*, corresponding to gray box in *Figure 1C*). The multi-resolution imaging allowed us to analyze the distribution of apical and basal surfaces of single hepatocytes (*Figure 1F*), as well as sinusoidal and BC network geometry (*Figure 1E*) in relation to the CV-PV axis. Quantitative structural parameters are shown in *Figure 1—figure supplement 2* and *Figure 1—figure supplement 3*. Analysis of thousands individual hepatocytes revealed the full complexity of distribution of the apical, lateral and basal surfaces.

### Hepatocytes display biaxial cell polarity

The liver parenchyma appears to lack a regular structure at the mesoscopic scale (*Figure 1*). Yet, its functional requirements suggest the existence of hidden order. To reveal it, we examined the orientation of hepatocyte polarity in the tissue. In simple epithelia, such as in the kidney and intestine, apico-basal cell polarity can be described by a single vector pointing from the cell center to a single apical pole (*Bryant and Mostov, 2008*; *Marcinkevicius et al., 2009*; *Treyer and Müsch, 2013*) (see schematic in *Figure 2A*). However, hepatocyte polarity cannot be described by a single apico-basal polarity axis.

To describe hepatocyte polarity, we used nematic axes. A nematic axis could be informally introduced as a two-headed vector, in contrast with the one-headed vector commonly used to describe vectorial cell polarity. Nematic axes have been used to describe anisotropic structures in physics, for example liquid crystals (*Gennes and Prost, 1995*). Liquid crystals are composed of anisotropic units such as elongated molecules, which can be partially oriented along a common axis. The individual units display variations in orientation, yet on average exhibit overall order, termed nematic order. Nematic order is a state of matter intermediate between perfect crystals and amorphous liquids.

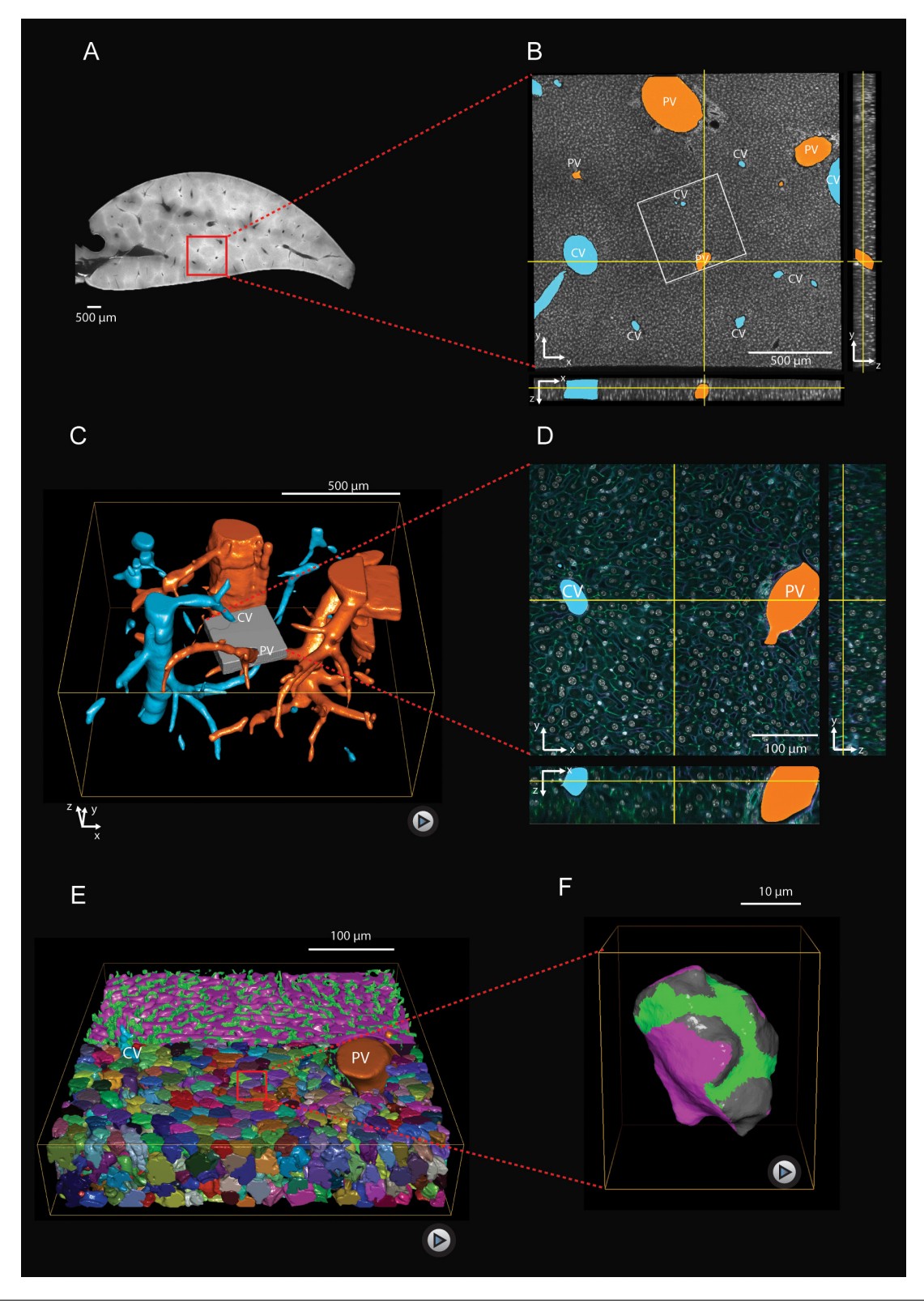

**Figure 1.** Multi-resolution imaging and 3D reconstruction of the mouse liver lobule. (A, B) Low-resolution imaging of an optically cleared liver tissue slice, stained for hepatocyte cell borders (cyan, Phalloidin) and nuclei (gray, DAPI); voxel size 1 μm x 1 μm x 1 μm. Central veins (CV, cyan) and portal veins (PV, orange) are highlighted. (C) 3D reconstruction from a stack of low-resolution images from 10 serial slices (*Figure 1—video 1*). (D) High-resolution imaging was performed in a sub-region (indicated as gray box in panel (C) and stained with four different markers for hepatocyte cell
*Figure 1 continued on next page*

*Figure 1 continued*

borders (cyan, Phalloidin), nuclei (gray, DAPI), hepatocyte apical plasma membrane (green, CD13), and basal plasma membrane (magenta, fibronectin/laminin) – facing the sinusoidal endothelial cells (*Figure 1—figure supplement 1*); voxel size 0.3 μm x 0.3 μm x 0.3 μm. (E) Reconstruction of sinusoidal (magenta) and bile canaliculi (green) networks connecting CV and PV, as well as contacting hepatocytes (*Figure 1—video 2*). (F) 3D representation of a single hepatocyte showing apical (green), basal (magenta) and lateral (gray) plasma membrane domains (*Figure 1—video 3*). A quantitative analysis of the structural parameters of hepatocytes and the networks (BC and sinusoids) along the CV-PV axis is shown in *Figure 1—figure supplement 2* and *Figure 1—figure supplement 3*, respectively.

DOI: https://doi.org/10.7554/eLife.44860.002

The following video, source data, and figure supplements are available for figure 1:

**Figure supplement 1.** Colocalization of sinusoids and basal plasma membrane.
DOI: https://doi.org/10.7554/eLife.44860.003

**Figure supplement 2.** Quantitative structural parameters of hepatocytes along CV-PV axis.
DOI: https://doi.org/10.7554/eLife.44860.004

**Figure supplement 2—source data 1.** Raw data structural parameters of hepatocytes along CV-PV axis.
DOI: https://doi.org/10.7554/eLife.44860.005

**Figure supplement 3.** Quantitative structural parameters of sinusoidal and BC networks along CV-PV axis.
DOI: https://doi.org/10.7554/eLife.44860.006

**Figure supplement 3—source data 1.** Raw data structural parameters of sinusoidal and BC networks along CV-PV axis.
DOI: https://doi.org/10.7554/eLife.44860.007

**Figure 1—video 1.** Supplementary video for *Figure 1C*. Tissue-level reconstruction of the liver lobule.
DOI: https://doi.org/10.7554/eLife.44860.008

**Figure 1—video 2.** Supplementary video for *Figure 1E*.
DOI: https://doi.org/10.7554/eLife.44860.009

**Figure 1—video 3.** Supplementary video for *Figure 1F*.
DOI: https://doi.org/10.7554/eLife.44860.010

The concept of nematic order was used to describe for example liquid crystal displays (LCD) in engineering, and alignment of cell shape anisotropy in 2D epithelial tissues in biology (*Saw et al., 2017*). We asked whether tissue-scale nematic order could exist also in 3D tissues.

To characterize the complex 3D apico-basal polarity of hepatocytes, we introduced a new concept of biaxial cell polarity. To this end, we used not just one, but two nematic axes (see Materials and methods). Mathematically, these nematic axes are defined via a nematic tensor associated with each hepatocyte, which is characterized by two principal axes (a third axis can be deduced from the other two). The geometric meaning of these axes, here termed the bipolar and the ring axis, is best exemplified in two extreme cases (*Figure 2B, C*). In the bipolar case, a marker is concentrated on two opposite poles (*Figure 2B*), whereas in the ring case, the marker forms a belt around the cell (*Figure 2C*). In the first case, the bipolar axis passes through the two poles (orange axis, $\mathbf{a}_1$), whereas the ring axis is not uniquely defined (it is degenerate in the plane perpendicular to the bipolar axis). In the second case, the ring axis is perpendicular to the plane of the ring and well defined (cyan axis, $\mathbf{a}_2$), whereas the bipolar axis is degenerate. In the case of hepatocytes, the distribution of the apical plasma membrane is in between these two extreme cases, resulting in two well-defined perpendicular axes (*Figure 2D*). Each of the two cell polarity axes has an associated weight deduced from the nematic tensors (see Materials and methods), $\sigma_1$ for the bipolar axis and $\sigma_2$ for the ring axis (*Figure 2E*). The distribution of weights is skewed in favor of the belt-like apical surfaces. However, extreme cases described only by a single axis are very rare in the population of hepatocytes. We can define an analogous pair of axes for the distribution of basal plasma membrane, $\mathbf{b}_1$ and $\mathbf{b}_2$, yielding similar results (*Figure 2—figure supplement 1*). Therefore, the polarity of hepatocytes is characterized by two nematic tensors and four axes ($\mathbf{a}_1, \mathbf{a}_2, \mathbf{b}_1, \mathbf{b}_2$). Next, we explored the relationship between apical and basal biaxial cell polarity. We found a preferential parallel alignment for apical and basal axes of different types, i.e., $\mathbf{a}_1$ aligned with $\mathbf{b}_2$, and $\mathbf{a}_2$ with $\mathbf{b}_1$ (*Figure 2F*). In contrast, the apical and basal axes of the same type ($\mathbf{a}_1$ with $\mathbf{b}_1$ and $\mathbf{a}_2$ with $\mathbf{b}_2$) have preferentially a perpendicular orientation. This anti-correlation corroborates the mutual repulsion between apical and basal surfaces of hepatocytes. Moreover, we only observed weak correlations between cell shape and apical

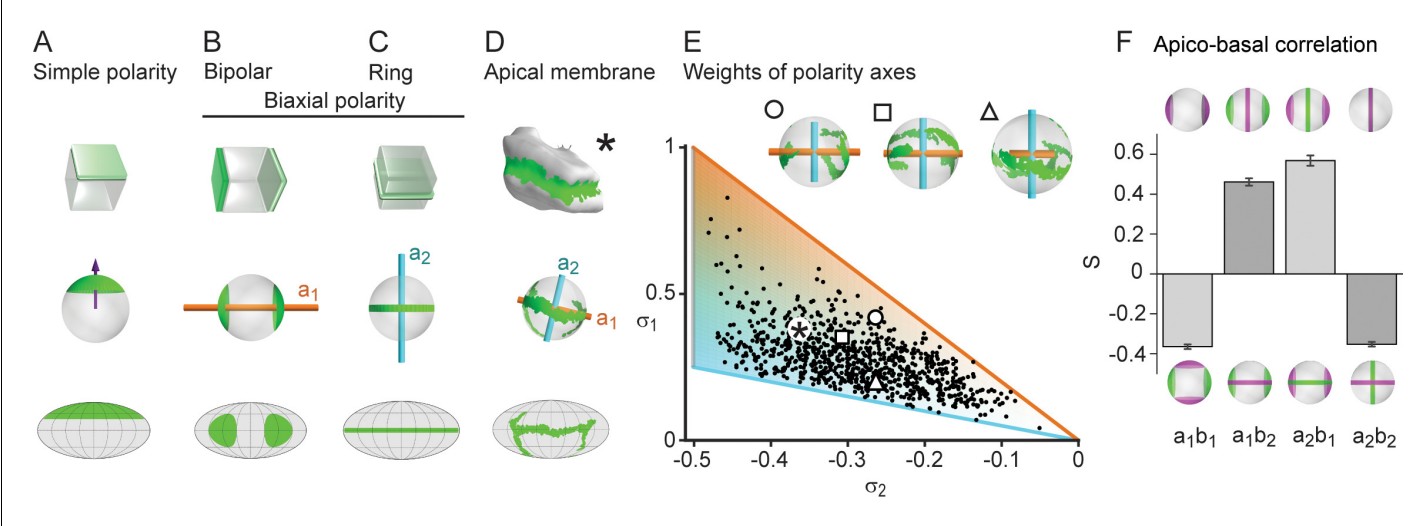

**Figure 2.** Biaxial cell polarity of hepatocytes. (**A**) Idealized representation of simple cell polarity, as found in cells of sheet-like epithelial tissue, showing schematic representation, spherical projection and Mollweide cartographic projection (top to bottom). Simple cell polarity is characterized by a single domain of apical membrane localized at one side of the cell, thus defining a vector (magenta arrow) that points toward the patch of apical plasma membrane. (**B, C**) Two extreme cases of biaxial polarity. Biaxial polarity, as introduced here, associates two nematic axes to complex membrane patterns: bipolar and ring axis. We show the bipolar axis ($a_1$, gold) for the idealized case of two antipodal poles of apical plasma membrane (pure bipolar polarity, ring axis degenerated), and the ring axis ($a_2$, cyan) for the case of a perfect ring of apical plasma membrane (pure ring polarity, bipolar axis degenerated). (**D**) Reconstructed 3D shape of typical hepatocyte with patches of apical plasma membrane (green). The spherical projection is characterized by well-defined bipolar and ring axes. (**E**) Respective weights of bipolar axis ($\sigma_1$) and ring axis ($\sigma_2$) for n=857 reconstructed hepatocytes, defined in terms of the eigenvalues of the nematic cell polarity tensor. Extreme cases of pure bipolar or pure ring polarity as shown in B and C correspond to the golden and blue line, respectively. Inset: Spherical projections for three example hepatocytes with corresponding polarity axes (indicated by symbols in scatter plot, corresponding to panel 2A). The analogous pair of axes for the distribution of basal plasma membrane showed similar results (*Figure 2—figure supplement 1*). (**F**) Cross-correlation analysis of nematic cell polarity axes for apical and basal plasma membrane patterns reveals that axes of same type are preferentially perpendicular, while axes of different type are preferentially parallel, indicating repulsion between apical and basal plasma membrane domains (n=3 animals). Error bars show standard deviations (s.d.).

DOI: https://doi.org/10.7554/eLife.44860.011

The following figure supplements are available for figure 2:

**Figure supplement 1.** Biaxial cell polarity of basal plasma membrane distribution.

DOI: https://doi.org/10.7554/eLife.44860.012

**Figure supplement 2.** Hepatocyte shape anisotropy.

DOI: https://doi.org/10.7554/eLife.44860.013

cell polarity axes, implying that these cellular features are independent to a large extent (*Figure 2— figure supplement 2*).

## Spatial patterns of cell polarity reveal liquid-crystal order in the liver lobule

We next examined the orientation of apical bipolar axes for all hepatocytes within a liver lobule between CV (cyan) and PV (orange) (*Figure 3A*). To highlight possible patterns of orientation, we performed local averaging of the bipolar axes over a width of 20 μm, corresponding to approximately one hepatocyte diameter (*Figure 3B*, see Materials and methods). This procedure revealed a spatial organization of hepatocyte polarity with a pattern of the bipolar axis oriented along lines that connect CV and PV. This pattern is reminiscent of flux lines generated by diffusive transport between CV and PV. Indeed, the stationary solution of the diffusion equation with source and sink on CV and PV, respectively, generates a pattern of flux (**J**) (*Figure 3C*) similar to the pattern of averaged bipolar apical axes (*Figure 3B*). A comparison of averaged apical bipolar axes and the flux pattern **J** is shown in color-code in *Figure 3D* and quantified in *Figure 3G*, where red color indicates strong alignment with the reference direction and blue denotes a perpendicular orientation. Performing the

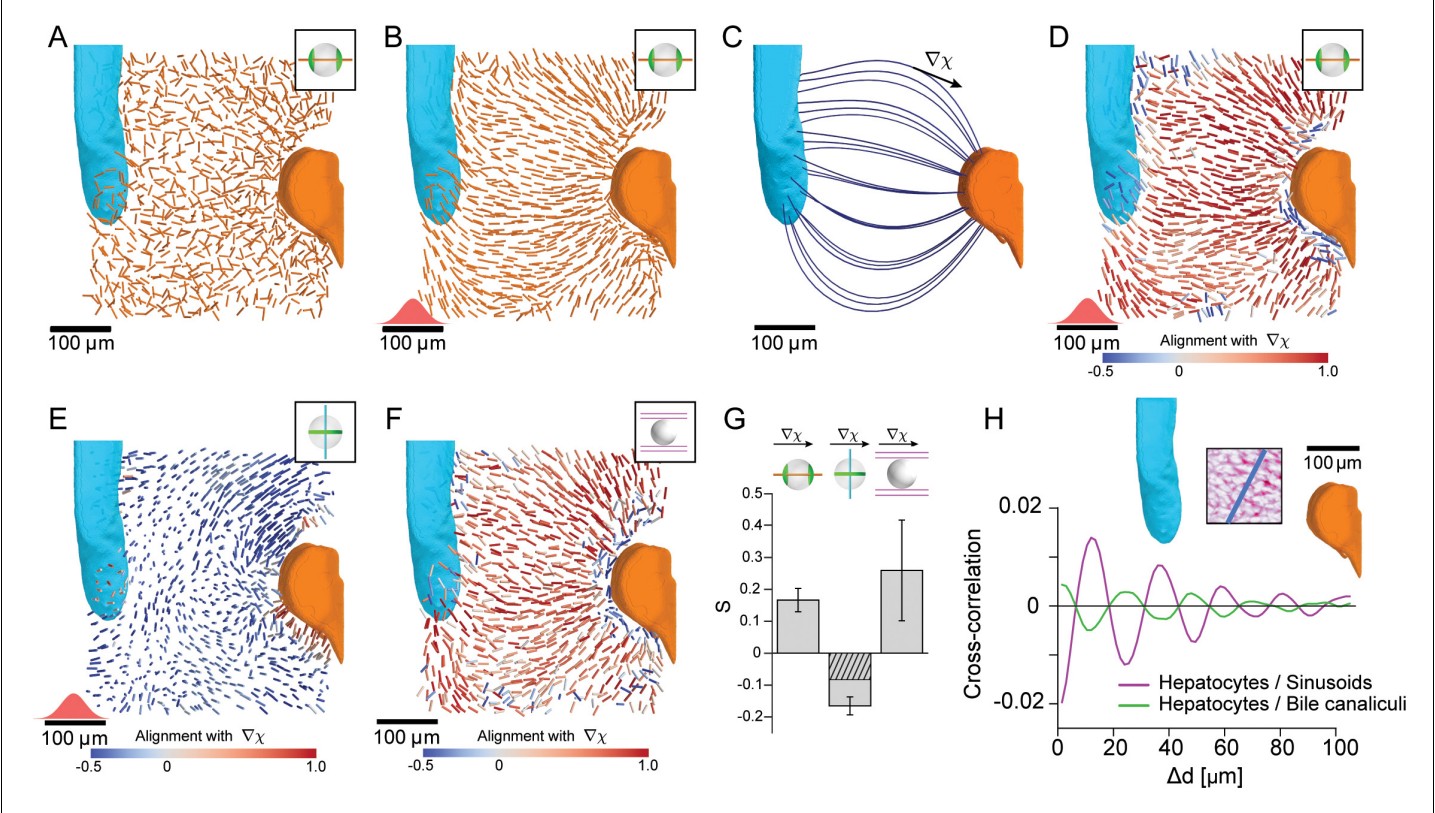

**Figure 3.** Lobule-level organization of nematic cell polarity. (A) Bipolar cell polarity axes of apical plasma membrane distribution ($a_1$) shown as lines of constant length for individual hepatocytes at their respective position in the lobule. (B) Same as A after local averaging using a 3D Gaussian kernel at each hepatocyte cell center (standard deviation 20 μm, indicated in red above the scale bar). (C) Lobule-level reference system with local reference direction (**J**) tangent to flow lines (blue), obtained by solving the diffusion equation with sources and sinks placed at the surface of PV (orange) and CV (cyan). (D) Same as B, but now axes are color-coded according to their alignment with the local reference direction (**J** defined in panel C). Red colors indicate parallel alignment, whereas blue indicate perpendicular co-orientation. (E) Same as D, but for the ring axis of apical plasma membrane distribution ($a_2$). (F) Same as D, but for the preferred direction of the local sinusoidal network surrounding each hepatocyte. (G) Quantification of alignment with local reference direction for apical bipolar axis, apical ring axis, and preferred sinusoid orientation. The correlation for the apical ring axis exceeds a trivial baseline (hatched bar) that follows from the correlation of the bipolar axis (see Materials and methods for details) (error bars denote standard deviations, n=3 animals). Analogous results for the preferred BC orientation are shown in *Figure 3—figure supplement 1*. (H) Layered order in the liver lobule. Upper: density of sinusoids in region-of-interest (average density projection along z-axis) and reference direction (blue). Lower: cross-correlation along reference direction between projected density of hepatocytes and sinusoids (magenta) and hepatocytes and bile canaliculi (green). The oscillatory signals reveal layered order with a wavelength of approximately one hepatocyte diameter. Detailed description in *Figure 3—figure supplement 2*. All scale bars 100 μm.

DOI: https://doi.org/10.7554/eLife.44860.014

The following figure supplements are available for figure 3:

**Figure supplement 1.** Anisotropy of BC network.

DOI: https://doi.org/10.7554/eLife.44860.015

**Figure supplement 2.** Control for layered order in the liver lobule.

DOI: https://doi.org/10.7554/eLife.44860.016

same analysis for the ring-like axis yielded a preferentially perpendicular orientation with respect to the reference direction **J** (*Figure 3E, G*). Consequently, belt-like apical domains are oriented to facilitate bile transport along the reference direction (see also *Figure 3—figure supplement 1*). The simultaneous alignment of two axes, bipolar and ring axis, is indicative of biaxial nematic order (*Luckhurst, 2015*). Next, we tested whether the observed order is truly biaxial. If the order were uniaxial (the null hypothesis), the correlation between the ring axis $a_2$ and the reference

direction **J** (*Figure 3G*, second bar) could be predicted from the alignment of the bipolar axis $a_1$ and **J** (*Figure 3G*, first bar), using the fact that $a_1$ and $a_2$ are perpendicular (see Materials and methods). However, we found that the alignment of the ring axis was significantly above the prediction (*Figure 3G*, hatched bar; p=0.014). This suggests the presence of biaxial order, which is confirmed by a detailed mathematical characterization in terms of biaxial order parameters described in *Scholich et al. (2019)*.

Considering that the sinusoidal network connects PV to CV to ensure blood flow, we expected this network to also exhibit patterns of orientation along the CV-PV axis. To test for this, we determined a local anisotropy axis of the sinusoidal network (see Materials and methods) and found significant parallel alignment between this anisotropy axis and **J**, see *Figure 3F, G*. Our analysis shows that cell polarity axes of hepatocytes and the anisotropy of the sinusoidal network display biaxial nematic order in the liver lobule. In soft condensed matter physics, such organization is known to result from either weak interactions between anisotropic units, or interactions with an external field, thus creating a liquid crystal-type of organization (*Gennes and Prost, 1995*). Liquid-crystal order, as found in displays of electronic devices, is characterized by orientational order of basic units, for example approximately parallel alignment, yet lack of the translational order of crystalline packings common of solids.

Next, we investigated if translational order could be found at least in one spatial direction. For this, we calculated the cross-correlation for density variations of sinusoids, hepatocytes and BC along a direction perpendicular to the CV-PV axis (see scheme in *Figure 3H*). The cross-correlations revealed a periodicity of structures along this direction with a characteristic length-scale of 24 μm. This periodicity approximately equals the sum of the hepatocyte and sinusoid tube diameters. In contrast, we found no evidence of periodic structures in the direction of the CV-PV axis (see *Figure 3—figure supplement 2*). Such periodicity in only one direction is a hallmark of a layered structure. Therefore, sinusoids, hepatocytes, and BC exhibit a layered organization, with most hepatocytes forming a single layer sandwiched between sinusoidal cells, supporting earlier structural models (*Elias and Bengelsdorf, 1952*; *Elias, 1949b*; *Elias, 1949c*).

## Bidirectional communication between hepatocytes and sinusoids is necessary for the maintenance of tissue structure

The layered organization of sinusoids and hepatocytes prompts the question of how coordination between these two cell types is achieved. It has been proposed that the sinusoidal endothelial cells are the main organizers by self-ordering and enforcing the position and, hence, the polarity of hepatocytes (*Hoehme et al., 2010*; *Sakaguchi et al., 2008*). This requires the establishment of apicobasal polarity with the basal plasma membrane of hepatocytes facing the sinusoidal endothelial cells. A candidate pathway orienting the basal surface of hepatocytes is that of the transmembrane ECM receptors Integrins (*Yu et al., 2005*). Perturbation of this pathway should result in a flawed coordination between sinusoids and hepatocytes and, consequently, defects in the liquid-crystal order of hepatocyte polarity. Furthermore, if the communication between sinusoids and hepatocytes were unidirectional, the sinusoidal network would be predicted to remain unaltered. To test these predictions, we silenced Integrin-β1 in the liver lobule *in vivo*. The injection of siRNAs formulated into lipid nanoparticles (LNP) provides the advantage of silencing genes with high efficacy and specificity in hepatocytes of adult mice (*Bogorad et al., 2014*; *Zeigerer et al., 2012*). Using super-resolution microscopy, we verified that Integrin-β1 expression was ablated in both the lateral and basal plasma membrane of hepatocytes but not in the sinusoidal endothelial cells (*Figure 4—figure supplement 1*). Injection of siRNAs against Integrin-β1 resulted in a 90% reduction in expression in comparison with control (injected with LNP-formulated Luciferase siRNA), as previously described (*Bogorad et al., 2014*). Loss of Integrin-β1 in liver parenchymal cells led to barely detectable alterations during the first 2–4 weeks. However, after 7 weeks of treatment with Integrin-β1-specific but not control siRNAs, when a significant number of hepatocytes are naturally replaced (*Magami et al., 2002*), we detected major alterations in liver tissue organization.

First, the BC network appeared disrupted and more branched (due to an increase of dead-end branches) (*Figure 4A*). Interestingly, the biaxial cell polarity of hepatocytes was not compromised. We observed almost unchanged correlation patterns between apical and basal cell polarity axes (*Figure 4B*) and indistinguishable distributions of their weights (*Figure 2—figure supplement 1*). This suggests that hepatocyte polarity is maintained by a cell-autonomous mechanism. On the scale

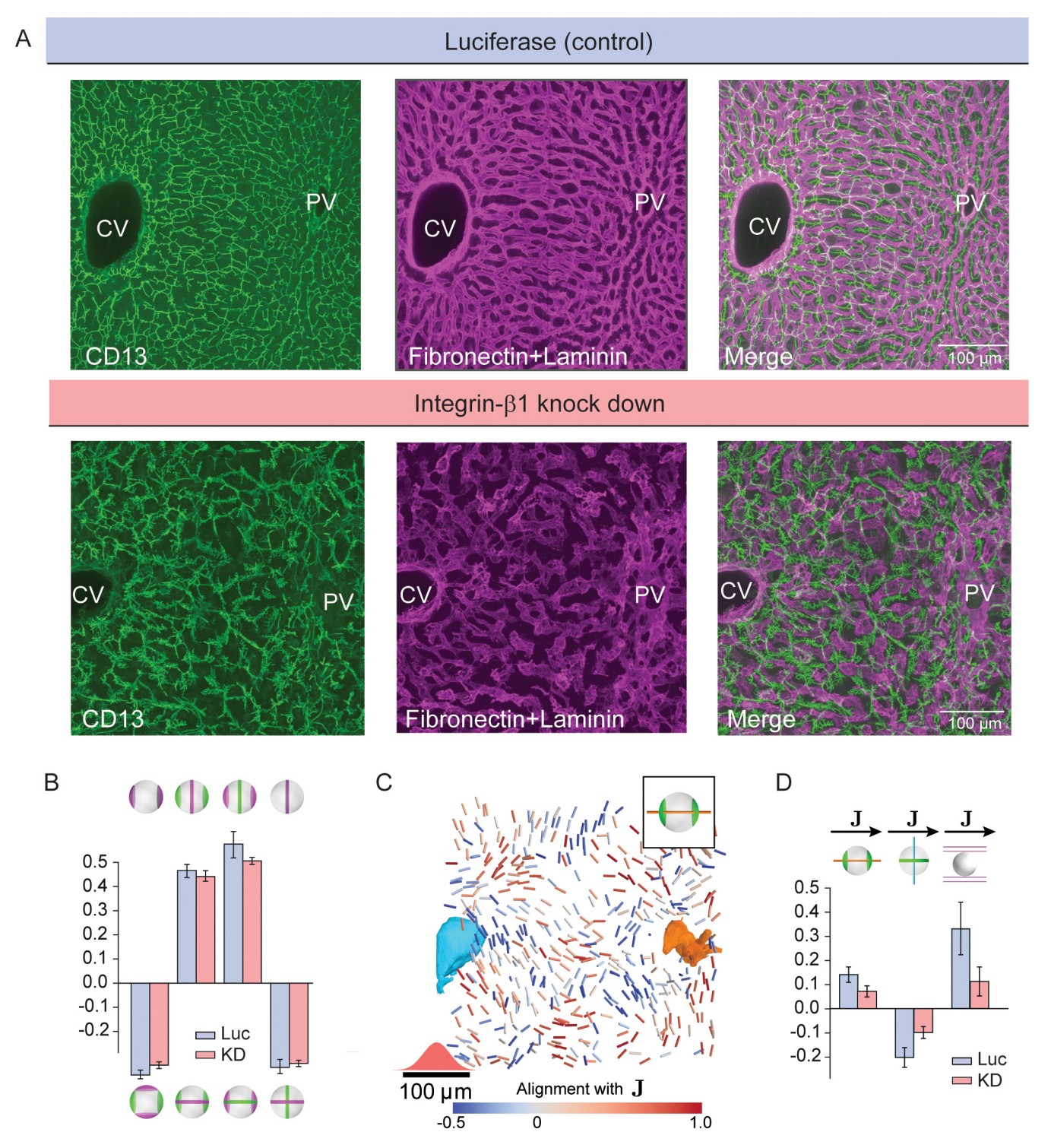

**Figure 4.** Liquid-crystal order, but not biaxial polarity of hepatocytes, is perturbed in Integrin-β1 KD mice. (**A**) Silencing Integrin-β1 in the liver results in distortion of both bile canalicular and sinusoidal networks, with reduced apparent alignment with the CV-PV axis in comparison to control conditions. Shown are representative samples for control conditions -upper panels, siRNA against Luciferase (Luc) and Integrin-β1 knock down -lower panels, siRNA against Integrin-β1 receptor (KD) stained for bile canalicular network (left, CD13 staining), sinusoidal network (middle, fibronectin/laminin staining), and merge (right). All panels correspond to maximal intensity z-projection of 60 μm of liver tissue. (**B**) Individual hepatocytes retain their biaxial cell polarity in Integrin-β1 KD, as revealed by cross-correlation analysis of nematic cell polarity axes for apical and basal plasma membrane patterns (analogous to *Figure 2F*). (**C, D**) In contrast, the alignment of biaxial cell polarity axes and the local preferred direction of the sinusoidal network with **J** are reduced in

*Figure 4 continued on next page*

*Figure 4 continued*

Integrin-β1 KD. Panel C shows bipolar cell polarity axes of apical plasma membrane color-coded according to their alignment with the local reference direction (**J**) (analogous to *Figure 3D*). Panel D shows the quantification of alignment of apical bipolar axis, apical ring axis, and preferred sinusoid orientation with local reference direction (analogous to *Figure 3G*). A detailed graphical representation can be found in *Figure 4—figure supplement 1*. Statistics in B, D: mean+/-s.d. for n=5 animals (Luc) and n=4 animals (KD); statistical significance: panel B: $\mathbf{a}_1 - \mathbf{b}_1$:$p = 0.015$, $\mathbf{a}_1 - \mathbf{b}_2$:$p = 0.277$, $\mathbf{a}_2 - \mathbf{b}_1$:$p = 0.118$, $\mathbf{a}_2 - \mathbf{b}_2$:$p = 0.410$; panel D: $\mathbf{a}_1 - \mathbf{J}$:$p = 0.007$, $a_2 - \mathbf{J}$:$p = 0.002$, $sinusoid - \mathbf{J}$:$p<0.008$, two-sided t-test assuming unequal variances).

DOI: https://doi.org/10.7554/eLife.44860.017

The following figure supplements are available for figure 4:

**Figure supplement 1.** Analysis of the Integrin-β1 expression by immunofluorescence staining.

DOI: https://doi.org/10.7554/eLife.44860.018

**Figure supplement 2.** Disturbed nematic liquid-crystal order in Integrin-β1 knock-down.

DOI: https://doi.org/10.7554/eLife.44860.019

of individual hepatocytes, the only change was a significant increase in apical surface at the expense of the basal surface (*Figure 1—figure supplement 2*). However, the long-range order of hepatocyte cell polarity was strongly perturbed (*Figure 4C, D* and *Figure 4—figure supplement 2*). Surprisingly, despite the silencing of Integrin-β1 being limited to hepatocytes at the used dosage, the sinusoidal network was also severely disrupted, with loss of its long-range organization (*Figure 4A, D*). This suggests that also hepatocytes provide instructions to sinusoidal endothelial cells. In conclusion, Integrin-β1 KD results in loss of long-range liquid-crystal order of the liver lobule and a perturbed coordination between BC and sinusoidal networks.

## Discussion

Determining the structure of a protein, that is the three-dimensional arrangement of amino acids, allows making predictions on its function, intra- and inter-molecular interactions, as well as mechanisms of action and mutations that could alter its activity. Similarly, elucidating the structure of a tissue allows making predictions on how cells interact with each other and self-organize to form a functional tissue, including molecular mechanisms governing these processes (*Hunter and de Bono, 2014*). While some progress has been made in understanding 2D tissues (*Dye et al., 2017*; *Etournay et al., 2016*; *Hirst and Charras, 2017*; *Legoff et al., 2013*; *Marcinkevicius et al., 2009*; *Saw et al., 2017*; *Saw et al., 2018*; *Zallen, 2007*) such as simple epithelia, the architecture of 3D tissues and its relation to function are poorly understood. The liver exemplifies this problem. Seventy years ago, Hans Elias pioneered an idealized structural model of liver tissue based on a crystalline order of cells (*Elias, 1949b*; *Elias, 1949c*). Although his model captured some essential features of liver architecture, it could not explain the heterogeneity of cells and the amorphous appearance of the tissue.

In this study, we discovered novel design principles of liver tissue organization. We found that hepatocytes, BC and sinusoidal networks are organized as a layered structure, with a spacing of about one hepatocyte diameter and orientation along the PV-CV axis, consistent with Elias' model of hepatic plates. However, a breakthrough from our analysis was that, by using biaxial nematic tensors to describe hepatocyte polarity, we discovered that the polarity axes of individual hepatocytes are not random but display a liquid-crystal order on the scale of the lobule.

It has been proposed that the sinusoidal network forms a scaffold structure that guides hepatocyte polarity and BC network organization (*Hoehme et al., 2010*; *Sakaguchi et al., 2008*). We propose an alternative organizational principle based on hierarchical levels of structural order (*Figure 5A*). At the cellular level, hepatocytes display biaxial cell polarity of apical membrane distribution, distinct from the polarity in simple epithelia. At the multi-cellular level, the apical polarity axes of hepatocytes and the preferred direction of the sinusoidal network are aligned. Hepatocytes, BC and sinusoids exhibit a layered organization, where the layers are parallel to the veins. On the lobule level, we observed liquid-crystal order of hepatocyte polarity. This represents an intermediate state of order between highly ordered crystals and disordered liquids (*Figure 5B*). The hierarchy of structural order could conceivably be explained by local rules of cell-cell communication in

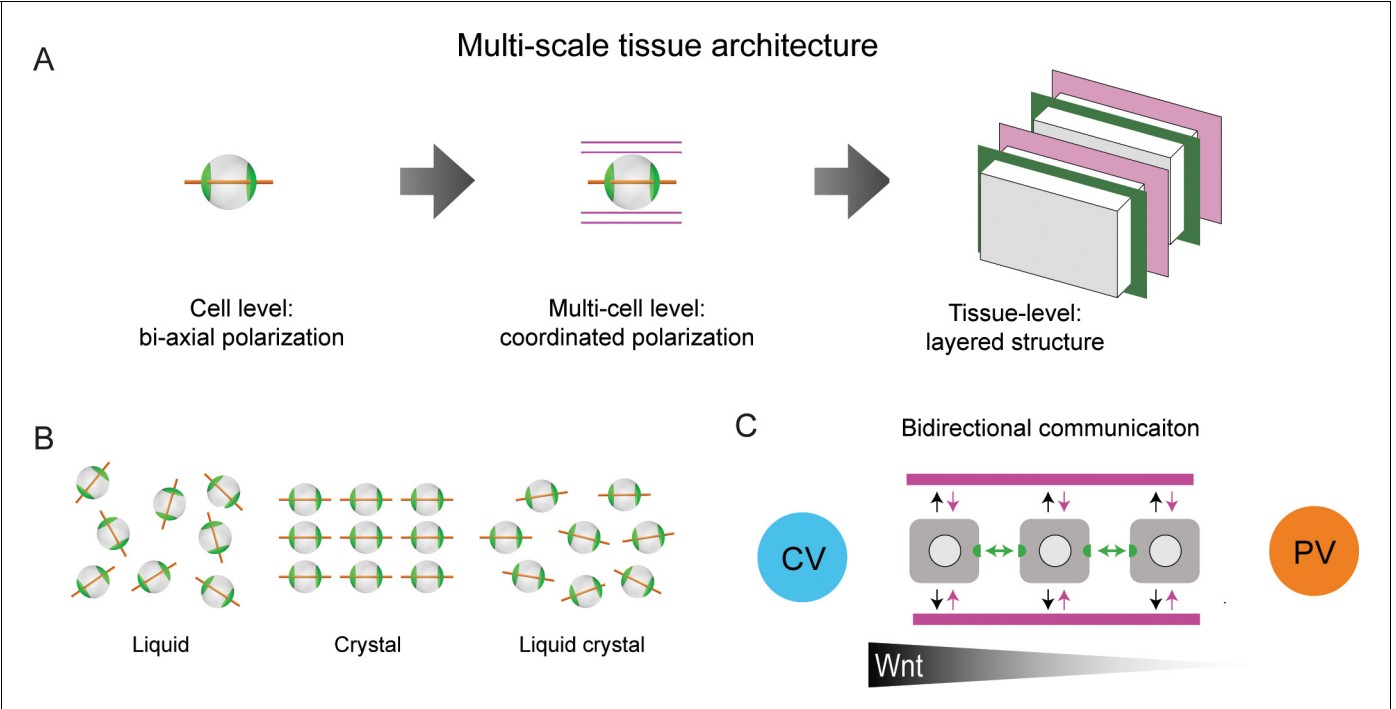

**Figure 5.** Proposed model of liver tissue architecture. (A) Our work proposes a new multi-scale model of liver architecture, characterized by liquid-crystal order of hepatocytes with biaxial nematic cell polarity, co-alignment of hepatocyte polarity and preferred direction of the sinusoidal network, and layered order of alternating sinusoidal network and layers of hepatocytes with a thickness of one cell diameter. (B) Cartoon representation of isotropic liquid with lack of positional and orientational order, crystalline order, and nematic liquid-crystal with orientational order. (C) Schematic of bidirectional communication between sinusoids and hepatocytes. Long-range gradients, for example Wnt signaling, could provide alignment cues for the orientation biaxial cell polarity of hepatocytes (top-down organization), in addition to local interactions between hepatocytes and sinusoids (bottom-up organization).

DOI: https://doi.org/10.7554/eLife.44860.020

combination with global cues (e.g. morphogen gradients). Silencing Integrin-β1 provides a clue into the molecular mechanism underlying the local communication between hepatocytes and sinusoids. We found that the biaxial cell polarity of individual hepatocytes was maintained. In contrast, the liquid-crystal order was perturbed, that is tissue-scale alignment of cell polarity, and sinusoidal network anisotropy. Sinusoid-hepatocyte co-alignment could result from a self-organization mechanism, whereby sinusoids guide hepatocyte polarization, while hepatocytes provide instructive signals that guide sinusoidal network organization (*Figure 5C*). This bidirectional communication provides a mechanism of self-organization for tissue development and maintenance. This points at a novel role of Integrin-β1 in orchestrating tissue structure by coupling the anisotropy of the sinusoidal network with the orientation of hepatocyte polarity, with boundary conditions set by the large veins.

In the liver, sinusoids and BC each form a single connected network. The two networks must be spatially separated, but also contact every hepatocyte to maximize the efficiency of fluid transport in the tissue. Furthermore, the networks are not tree-like, but highly interconnected and redundant with multiple loops and multiple contacts to each hepatocyte. This architecture confers robustness against local damage and failure. Our structural model of liquid-crystal order provides an explanation of how cells self-organize through local interactions to achieve this particular architecture, which is compatible with continuous homeostatic remodeling of the tissue.

Our model of liver tissue organization defines the road ahead. For example, are there intermediate (multi-cellular) units of structural organization between the cell and tissue level? Our study provides a general framework for elucidating the rules of cell-cell interactions and structural order of 3D tissues beyond the liver.

# Materials and methods

**Key resources table**

| Reagent type (species) or resource | Designation | Source or reference | Identifiers | Additional information |
|---|---|---|---|---|
| Antibody | anti-Flk-1 (goat polyclonal) | R and D system | AF644 / RRID:AB_355500 | (1:200) |
| Antibody | anti-laminin (rabbit polyclonal) | Sigma | L9393/RRID:AB_477163 | (1:5000) |
| Antibody | anti-fibronectin (rabbit polyclonal) | Millipore | AB2033/RRID:AB_2105702 | (1:1000) |
| Antibody | anti-CD13 (rat monoclonal) | Novus | NB100−64843/RRID:AB_959651 | (1:500) |
| Antibody | anti-integrin ß1 (rat monoclonal) | Millipore | MAB1997/RRID:AB_2128202 | (1:1000) |
| Antibody | Donkey anti-goat Alexa Fluor 647 | Invitrogen | A21447/RRID:AB_2535864 | (1:1000) |
| Antibody | Donkey anti-rabbit Alexa Fluor 647 | Invitrogen | A31573/RRID:AB_2536183 | (1:1000) |
| Antibody | Donkey anti-rat CF 568 | Biotium | 20092/RRID:AB_10559037 | (1:1000) |
| Other | Phalloidin-488 | LIFE technologies | A12379/RRID:AB_2315147 | (1:150) |
| Other | Dapi | LIFE technologies | D1306/RRID:AB_2629482 | (1 µg/ml) |
| Sequence-based reagent | LNP-formulated siRNAs against luciferase | *Bogorad et al., 2014* | | (1 mgKg-1) |
| Sequence-based reagent | LNP-formulated siRNAs against Integrin-ß1 | *Bogorad et al., 2014* | | (1 mgKg-1) |
| Strain, strain background (*M. musculus* C57BL/6JOlaHsd) | Wild type, Luciferase, Integrin-ß1 knock down | Charles River Laboratory | | |
| Software, algorithm | MotionTracking | *Morales-Navarrete et al., 2015* | | |
| Software, algorithm | MuSiCal | *Agarwal and Macháň, 2016* | | |

## Mice

To study liver tissue organization under normal conditions, 6–9 weeks old C57BL/6JOlaHsd mice (two males and one female) were purchased from Charles River Laboratory. For the knocked-down experiments, eight weeks old male mice were purchased from the same company. Integrin-β1 was knocked-down in mice (four males) by injecting siRNA formulated into lipidoid-based nanoparticles (LNP) that primarily target hepatocytes. An analogous treatment using siRNA against luciferase was used as control (five males). A complete description of the knock-down experiments can be found in *Bogorad et al. (2014)*. All procedures were performed in compliance with German animal welfare legislation and in pathogen-free conditions in the animal facility of the MPI-CBG, Dresden, Germany. Protocols were approved by the Institutional Animal Welfare Officer.

## Sample collection, immunostaining and optical clearing

Mice livers were fixed through intracardiac perfusion with 4% paraformaldehyde and post-fixed overnight at 4˚C with the same solution. Eight to ten serial slices were cut in a vibratome (thickness 100 µm), corresponding to a total thickness of 800–1000 µm. For immunostaining, we used anti-CD13 (Novus, cat NB100-64843, rat, 1/500), anti-fibronectin (Millipore, cat AB2033, rabbit, 1/1000), anti-laminin (Sigma, cat L9393, rabbit, 1/5000), anti-Flk-1 (R and D system, cat AF644, goat, 1/200), phalloidin-488 (LIFE technologies, cat A12379, 1/150) and DAPI (LIFE technologies, cat D1306, 1 µg/ml). Liver slices were permeabilized with 0.5% Triton X-100 in PBS for 60 min at room temperature (RT).

Both, primary and secondary antibodies were diluted in TxBuffer (0.2% gelatin, 300 mM NaCl, 0.3% Triton X-100 in PBS) and each antibody was incubated with the tissue for 2 days at room temperature. For optical clearing, we used a modified version of SeeDB (*Ke et al., 2013*). The first day the liver slices were incubated consecutively in 25% fructose for 4 hr, 50% fructose for 4 hr and 75% fructose overnight. The second day the samples were transferred to 100% fructose (100% wt/v fructose, 0.5% 1-thioglycerol, 0.1M phosphate buffer pH7.5) and the third day we left the samples in SeeDB solution (80.2% wt/wt fructose, 0.5% 1-thioglycerol, 0.1M phosphate buffer pH7.5) until the images were acquired at the microscope. Different concentrations of fructose were prepared diluting 100% fructose with water.

### Imaging and image analysis

Liver samples were imaged in an upright multiphoton laser-scanning microscope (Zeiss LSM 780 NLO) equipped with Gallium arsenide phosphide (GaAsp) detectors. Liver slices were imaged twice at low (25x/0.8 Zeiss objective, 1 µm voxel size) and high resolution (63x/1.3 Zeiss objective, 0.3 µm voxel size), respectively. Low-resolution images were taken for the 3D reconstruction of big veins (central and portal veins) where the high resolution images were embedded. High-resolution images were acquired between selected central to portal vein (CV-PV) axes to resolve sub-cellular structures such as apical surfaces of hepatocytes. Both high- and low-resolution images were processed and segmented with the Motion Tracking software as described in *Morales-Navarrete et al. (2015)* and *Morales-Navarrete et al. (2016)*. The two-dimensional (2D) super-resolved images were generated by applying the Multiple Signal Classification Algorithm for super-resolution fluorescence microscopy (MuSiCal) (*Agarwal and Macháň, 2016*) to a set of 50 sequential time-lapse confocal images with 0.1 µm pixel size.

### Mollweide projection

We used the planar, pseudo-cylindrical Mollweide projection to visualize polarity patterns in 2D (*Figure 2*). The Mollweide projection preserves distances and areas at the expense of distorting shapes and is familiar from global maps of the Earth (*Steinwand et al., 1995*). For each cell, the apical plasma membrane domain of the cell surface was radially projected on a sphere placed at the volumetric center of the cell. To define a reference orientation for each cell, the bipolar axis of its projected basal plasma membrane ($b_1$) was chosen as the 'north pole-south pole' axis for projection while the bipolar axis of the apical plasma membrane ($a_1$) defined the position of the zero meridian, which is placed vertically in the center of the projection. This relates the 'poles' to the largest basal patches and the 'equator' to apical patches with the largest apical patch pointing towards the reader.

### Nematic tensors of polarity marker distribution of individual hepatocytes

In our software MotionTracking, the surface of a cell is represented as a triangulated mesh. For each vertex of the mesh, it is stored if this vertex was identified to belong to membrane patches rich in the apical polarity protein marker. Triangles are considered to belong to the apical plasma membrane patch if at least two vertices of the triangle have apical identity.

To compute the nematic tensor of apical polarity, triangles were first projected on a unit sphere to avoid distortions by non-spherical cell shapes, see *Figure 2D*. The center of the sphere is placed at the volumetric center of the cell. This projection assumes cell shapes to be star-convex with respect to the volumetric center of the cell. As a test, the sum of projected areas yielded $(4.035 \pm 0.086)\pi$, consistent with the surface area $4\pi$ of a unit sphere.

The nematic tensor of apical polarity is defined as a sum of all projected triangles with apical identify, indexed by $i \in I_{\mathrm{apical}}$

$$N_{\alpha\beta} = \frac{3}{2} \frac{1}{\sum_i A_i} \sum_{i \in I_{\mathrm{apical}}} A_i \left( n_\alpha^{(i)} n_\beta^{(i)} - \frac{1}{3}\delta_{\alpha\beta} \right). \tag{1}$$

Here, $A_i$ denotes the projected area of the projected triangle with index $i$ and $\mathbf{n}^{(i)}$ its surface normal vector. Einstein summation convention is assumed.

The rank-2 tensor $N_{\alpha\beta}$ is traceless and symmetric, and can thus be diagonalized with normalized eigenvectors $\mathbf{a}_1, \mathbf{a}_2, \mathbf{a}_3$, and respective eigenvalues $\sigma_1, \sigma_2, \sigma_3$. Without loss of generality, we assume $\sigma_2 \leq \sigma_3 \leq \sigma_1$. We refer to $\mathbf{a}_1$ as the bipolar axis, and $\mathbf{a}_2$ as the ring axis, and to $\sigma_1, \sigma_2$ as their respective weights. Note that the eigenvectors are only determined up to sign, and thus each specify an axis. The third axis can be deduced from the other two as $\mathbf{a}_3 = \pm \mathbf{a}_1 \times \mathbf{a}_2$, while $\sigma_3 = -\sigma_1 - \sigma_2$. We did not observe strong correlations between these weights of polarity axes and the alignment of the axes with the lobule-level reference field $\mathbf{J}$ defined below (not shown).

The definition of the nematic tensor of basal polarity and the respective polarity axes $\mathbf{b}_1$ and $\mathbf{b}_2$, is analogous.

Similarly, we define a preferred axis of the local sinusoidal network. The skeleton of the network is characterized by segments with respective vectors $\mathbf{n}^{(i)}$ and segment lengths $l^{(i)}$, for $i = 1, \ldots, N$. We define a nematic tensor that characterizes the anisotropy of the network

$$N_{\alpha\beta} = \frac{3}{2} \frac{1}{\sum_i l_i} \sum_{i=1}^{N} l_i \left( e_\alpha^{(i)} e_\beta^{(i)} - \frac{1}{3} \delta_{\alpha\beta} \right). \tag{2}$$

We refer to the axis parallel to the eigenvector corresponding to the largest eigenvalues as the sinusoid preferred axis.

## Local averaging of polarity axes

For *Figure 3B, D, E* and *Figure 4—figure supplement 1*, local averaging of polarity axes was performed. Specifically, given unit vectors $\mathbf{a}^{(j)}$ that characterize a cell polarity axis of given type for individual hepatocytes (located at estimated cell center positions $\mathbf{x}^{(j)}$ labeled by an index $j$, we computed nematic tensors

$$\bar{N}_{\alpha\beta}(\mathbf{x}) = \frac{3}{2} \sum_j w\left( |\mathbf{x}^{(j)} - \mathbf{x}| \right) \left( \mathbf{a}_\alpha^{(j)} \mathbf{a}_\beta^{(j)} - \frac{1}{3} \delta_{\alpha\beta} \right), \tag{3}$$

where $w(\mathbf{x})$ denotes a Gaussian kernel, with standard deviation chosen as 20 μm (approximately one hepatocyte diameter). In the respective plots, we displayed the principal axis with maximal eigenvalue of the weighted-mean tensors $\bar{N}_{\alpha\beta}(\mathbf{x}^{(i)})$ evaluated at the positions $\mathbf{x}^{(i)}$.

## Lobule-level reference system (J)

The lobule-level reference system is defined by the location of the large veins within the imaging volume in terms of flux lines of solutions of the Poisson equation. The Poisson equation describes diffusive transport between spatially separated sources and sinks. Equivalently, the same solutions can be interpreted as an electrostatic potential, where positive and negative point charges correspond to the sources and sinks, respectively, and the electrostatic potential corresponds to a steady-state concentration field established by diffusion.

Below, we use the terminology of the electrostatic problem, which has the formal benefit that negative values of the $\chi$ field have a direct physical interpretation as a negative potential, whereas in the interpretation of concentration fields a homogeneous constant $\chi_0$ has to be added to the $\chi$ field to ensure that concentrations $c = \chi + \chi_0$ are non-negative.

The location of the large vessels (portal vein and central vein) are given as triangulated meshes as calculated by MotionTracking. Using the electrostatic analogy, point charges are placed at the location $\mathbf{r}_i$ of the triangle centers with point charges of strength $q_i$ proportional to the relative area of the triangle with respect to the total area of the corresponding vessel

$$q_i = \pm \frac{A_i}{\sum_j A_j}. \tag{4}$$

Here, the sum extends over all triangles of either the PV or CV mesh representation, respectively. The sign of the point charge $q_i$ are opposite between the two vessel types, that is negative for portal vein and positive for central vein. This choice corresponds to a uniform charge surface density on the surfaces of the large vessels with total net charge equal to ±1, respectively.

A scalar field $\chi$ is calculated by superposition of the Green's functions of all the point charges on the veins

$$\chi = q_i \sum_i \frac{1}{|\mathbf{r} - \mathbf{r}_i|},$$

(5)

where the sum extends over all triangles of the PV and CV mesh representation. The value of this scalar field defines a positional value indicating location between PV and CV. Negative values indicate closeness to the portal vein, whereas positive values indicate closeness to the central vein. The gradient of this scalar field $\mathbf{J} = \nabla \chi$ gives a reference direction at all locations within the lobule. This field of reference directions is used to determine alignment of the nematic axes derived by the polarity of hepatocytes and the anisotropy of the sinusoidal and BC networks.

## Nematic alignment parameter *S*

We consider an ensemble of axes, represented by unit vectors $\mathbf{e}^{(i)}$, together with an ensemble of reference axes with unit vectors $\mathbf{g}^{(i)}$, where $i = 1, \ldots, N$. We define the nematic alignment parameter as

$$S = \frac{1}{N} \sum_{i=1}^{N} \frac{3}{2} \left( \mathbf{e}^{(i)} \cdot \mathbf{g}^{(i)} \right)^2 - \frac{1}{2}.$$

(6)

We have $S = 1$ for the case that each axis $\mathbf{e}^{(i)}$ is perfectly parallel to its respective reference axis $\mathbf{g}^{(i)}$, whereas $S = -\frac{1}{2}$ for perfectly perpendicular relative alignment. For an isotropic orientation between axes and references axes, $S = 0$. We apply this general definition to cell polarity axes of hepatocytes and the preferred anisotropy axis of the local sinusoidal network for $\mathbf{e}^{(i)}$, while the curvilinear reference field $\mathbf{g}^{(i)} = \mathbf{J}$, evaluated at each cell position, provides local reference axes.

Our definition of the nematic alignment parameter S generalizes the known nematic order parameter (*Gennes and Prost, 1995*; *Luckhurst, 2015*). Specifically, if the reference axis in our definition is chosen constant and equal to the nematic symmetry axis of the ensemble, both definitions agree.

## Trivial baseline for nematic alignment parameter *S* for biaxial objects

We note a special property of the nematic alignment parameter for an ensemble of biaxial objects, characterized by respective unit vectors $\mathbf{e}_1^{(i)}, \mathbf{e}_2^{(i)}, \mathbf{e}_3^{(i)}$, $i = 1, \ldots, N$. Let $S_1$, $S_2$, $S_3$ be the respective nematic alignment parameters for each of the three different axes. Then, the maximum-likelihood estimate for $S_2$, given a known value for $S_1$, reads

$$S_2^{ML} = -\frac{1}{2} S_1.$$

(7)

This relationship follows from $S_1 + S_2 + S_3 = 0$ and $S_2^{ML} = S_3^{ML}$. This trivial baseline is shown in *Figure 3G* as hatched bar for the correlation between the ring axis $\mathbf{a}_2$ and $\mathbf{J}$.

## Cross-correlation analysis

To investigate a possible layered order of liver tissue, we performed a cross-correlation analysis, see *Figure 3H*. We first selected a region of interest in the tissue sample, placed in the middle of the lobule. In this selected region, the field lines of the lobule-level reference field ($\mathbf{J}$) are approximately straight and parallel, which facilitated analysis. The region is shown as a square in the inset of *Figure 3H* (enlarged in *Figure 3—figure supplement 2A*). Inside the square region, the mean projected density of the sinusoidal network (projected along the z-axis) based on segmented voxelated data is shown. Next, we investigated layered order between the positions of hepatocytes and the sinusoidal and BC networks.

We calculated the cross-correlation between the mean projected density of the sinusoidal network and an analogously defined mean projected density of hepatocytes (*Figure 3—figure supplement 2B*). Specifically, we represent the mean projected density of the segmented sinusoidal network by a 2D pixel array $\mathbf{S}[n, m]$ and the averaged projected density of hepatocytes by a pixel array $\mathbf{H}[n, m]$, where $n$ and $m$ are the indices of the pixels in the 2D images. The normalized cross-correlation between the respective images was then calculated as

$$C_{\mathrm{SH}}[k,l] := \frac{1}{N_{\mathrm{S}} + N_{\mathrm{H}}} \sum_{n,m} \frac{1}{\sigma_{\mathrm{S}} \sigma_{\mathrm{H}}} (\mathbf{S}[n,\,m] - \mu_{\mathrm{S}})(\mathbf{H}[n+k,\,m+l] - \mu_{\mathrm{H}}), \tag{8}$$

where the sum runs over all pixels of $\mathbf{S}$. Here, $N_{\mathrm{S}}$ denotes the total number of pixels of $\mathbf{S}$, $\mu_{\mathrm{S}} = \sum_{n,m} \mathbf{S}[n,m]/N_f$ is the average of $\mathbf{S}$, and $\sigma_{\mathrm{S}}$ the standard deviation, defined as $\sigma_{\mathrm{S}} = \sqrt{\sum_{n,m} \left(\mathbf{S}[n,m] - \overline{\mathbf{S}}\right)^2 / N_f}$. Analogous definitions apply to $N_{\mathrm{H}}$, $\mu_{\mathrm{H}}$, and $\sigma_{\mathrm{H}}$. Outside its valid range, the array $\mathbf{H}[n,m]$ was zero-padded. The resultant 2D-cross correlation array $C_{\mathrm{SH}}[k,l]$ thus had twice the dimensions of $\mathbf{S}$.

We then computed the mean projection of this cross correlation array $C_{\mathrm{SH}}[k,l]$ on a line passing through $k = 0$, $l = 0$, and parallel to the blue reference line in the region of interest shown in the inset of *Figure 3H*. For this projection, we used a binning of 5 pixels, corresponding to a bin size of 1.5 µm. The use of a mean projection of 2D-cross correlation employed here instead of a conventional 1D-cross correlation along a single line reduced noise in the data and accounts for the fact of random phase shifts between neighboring layers.

An analogous cross correlation can be computed between the mean projected density of segmented hepatocytes and the BC network (*Figure 3—figure supplement 2C*). The final projected cross correlations between sinusoidal network and hepatocytes (magenta), as well as between sinusoidal network and bile canaliculi network (green) are shown in *Figure 3H* (reproduced in *Figure 3—figure supplement 2D*). As a control, *Figure 3—figure supplement 2E* shows the analogous plot for a control direction perpendicular to the blue reference direction).

## Acknowledgements

We thank Samuel Safran for stimulating discussions, and Stephan Grill, Peter Jansen, Carl Modes, Ivo Sbalzarini for a critical reading of the manuscript. We acknowledge generous allocation of computer time by the Center for Information Services and High Performance Computing (ZIH) at TU Dresden. We would like to thank the following Services and Facilities of the Max Planck Institute of Molecular Cell Biology and Genetics for their support: Biomedical Services (BMS) and Light Microscopy Facility (LMF). Funding: This research was financially supported by the European Research Council (ERC) (grant number 695646), the German Federal Ministry of Research and Education (BMBF) (LiSyM, grant number 031L0038 and SYSBIO II, grant No. 031L0044), the Deutsche Forschungsgemeinschaft (DFG) (Cluster of Excellence EXC 1056 cfaed), the Deutsche Forschungsgemeinschaft (DFG, German Research Foundation) under Germany's Excellence Strategy - EXC-2068 - 390729961- Cluster of Excellence Physics of Life of TU Dresden and the Max Planck Society (MPG).

## Additional information

### Competing interests

Frank Jülicher: Reviewing editor, *eLife*. The other authors declare that no competing interests exist.

### Funding

| Funder | Grant reference number | Author |
| --- | --- | --- |
| Bundesministerium für Bildung und Forschung | LiSyM-031L0038 | Fabián Segovia-Miranda Kirstin Meyer |
| Deutsche Forschungsgemeinschaft | Cluster of Excellence EXC 1056 cfaed | Benjamin M Friedrich |
| Max-Planck-Gesellschaft | Open-access funding | Hidenori Nonaka |
| Bundesministerium für Bildung und Forschung | SYSBIO II-031L0044 | Fabián Segovia-Miranda Kirstin Meyer |
| H2020 European Research Council | 695646 | Hernán Morales-Navarrete |

The funders had no role in study design, data collection and interpretation, or the decision to submit the work for publication.

## Author contributions
Hernán Morales-Navarrete, Conceptualization, Data curation, Software, Formal analysis, Validation, Investigation, Visualization, Methodology, Writing—original draft, Writing—review and editing; Hidenori Nonaka, Conceptualization, Validation, Investigation, Methodology; André Scholich, Conceptualization, Data curation, Software, Formal analysis, Investigation, Visualization, Methodology, Writing—original draft; Fabián Segovia-Miranda, Conceptualization, Validation, Investigation, Methodology, Writing—original draft, Writing—review and editing; Walter de Back, Formal analysis, Investigation, Writing—original draft; Kirstin Meyer, Investigation; Roman L Bogorad, Victor Koteliansky, Resources, Investigation; Lutz Brusch, Conceptualization, Supervision, Investigation, Writing—original draft; Yannis Kalaidzidis, Conceptualization, Software, Supervision, Investigation, Methodology, Writing—original draft, Writing—review and editing; Frank Jülicher, Conceptualization, Supervision, Funding acquisition, Methodology, Writing—original draft, Writing—review and editing; Benjamin M Friedrich, Conceptualization, Formal analysis, Supervision, Investigation, Visualization, Methodology, Writing—original draft, Project administration, Writing—review and editing; Marino Zerial, Conceptualization, Supervision, Funding acquisition, Investigation, Methodology, Writing—original draft, Project administration, Writing—review and editing

## Author ORCIDs
Hernán Morales-Navarrete https://orcid.org/0000-0002-9578-2556
André Scholich https://orcid.org/0000-0002-9393-5459
Fabián Segovia-Miranda https://orcid.org/0000-0003-1546-0475
Walter de Back http://orcid.org/0000-0003-4641-8472
Lutz Brusch http://orcid.org/0000-0003-0137-5106
Frank Jülicher https://orcid.org/0000-0003-4731-9185
Benjamin M Friedrich https://orcid.org/0000-0002-9742-6555
Marino Zerial https://orcid.org/0000-0002-7490-4235

## Ethics
Animal experimentation: All procedures were performed in compliance with German animal welfare legislation and in pathogen-free conditions in the animal facility of the MPI-CBG, Dresden, Germany. Protocols were approved by the Institutional Animal Welfare Officer (Tierschutzbeauftragter) and all necessary licenses were obtained from the regional Ethical Commission for Animal Experimentation of Dresden, Germany (Tierversuchskommission, Landesdirektion Dresden) (License number: DD24-5131/338/50).

## Decision letter and Author response
Decision letter https://doi.org/10.7554/eLife.44860.023
Author response https://doi.org/10.7554/eLife.44860.024

# Additional files

## Supplementary files
• Transparent reporting form
DOI: https://doi.org/10.7554/eLife.44860.021

## Data availability
All data generated or analysed during this study are included in the manuscript and supporting files. Source data files have been provided for Figure 1-figure supplement 2 and Figure 1-figure supplement 3.

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
