## [Decision Letter]

Thank you for submitting your article "Liquid-crystal organization of liver tissue" for consideration by *eLife*. Your article has been reviewed by three peer reviewers, and the evaluation has been overseen by a Reviewing Editor and Arup Chakraborty as the Senior Editor. The following individual involved in review of your submission has agreed to reveal their identity: Benoit Ladoux (Reviewer #2).

The reviewers have discussed the reviews with one another and the Reviewing Editor has drafted this decision to help you prepare a revised submission.

Summary:

The reviewers agree that this is a very interesting paper with several important contributions. The first is the use of state-of-the-art imaging to construct a 3D picture of the liver organisation. The second is the use of tools drawn for the physics of liquid crystal to show that the organisation of polarity domains at the surface of hepatocytes displays biaxial order with an apparent repulsion between apical and basal regions, and that there exists a nematic-like long range order of cell orientation at the tissue scale. The third is the finding that the hepatocyte polarity is maintained by a cell-autonomous mechanism, while bidirectional communication between hepatocytes and sinusoids is required to obtain the proper tissue organisation.

Essential revisions:

1) One concern regards the way the paper is organised and the way non-trivial notions of orientational order are introduced. It would be useful to define liquid crystal order earlier in the paper. The notion of biaxial polarity at the cell scale is a bit complex and it could be useful to discuss in general terms the concept of nematic order at the tissue scale first so that the need to define an orientation axis at the cell scale becomes more natural.

2) The nematic order is discussed in terms of apical-basal polarity. Does the cell shape reveal such an order as well? Could a nematic parameter be defined from the cell body elongation, and would it match the polarity axis? Is there a correlation between shape and the bipolar or ring axes?

3) The reference to a biaxial nematic order at the tissue scale (subsection “Spatial patterns of cell polarity reveal liquid-crystal order in the liver lobule”) is also unclear. This seems to suggest that one direction perpendicular to the flux orientation J is singled out, while this does not seem to be the case here.

4) Where is the evidence to show that the silencing of Integrin-β1 does not affect the sinusoidal cells? This is an important point to support the claim that the large-scale tissue organisation requires two-way communication between hepatocytes and endothelial cells.

---

## [Author Response]

Essential revisions:1) One concern regards the way the paper is organised and the way non-trivial notions of orientational order are introduced. It would be useful to define liquid crystal order earlier in the paper. The notion of biaxial polarity at the cell scale is a bit complex and it could be useful to discuss in general terms the concept of nematic order at the tissue scale first so that the need to define an orientation axis at the cell scale becomes more natural.

We agree with the comment and added a new paragraph to better introduce the concept of nematic cell polarity. (Results subsection “Hepatocytes display biaxial cell polarity”)

2) The nematic order is discussed in terms of apical-basal polarity. Does the cell shape reveal such an order as well? Could a nematic parameter be defined from the cell body elongation, and would it match the polarity axis? Is there a correlation between shape and the bipolar or ring axes?

Indeed, we had performed this analysis previously and have now included it in the manuscript. We defined cell shape axes in terms of a moment of inertia tensor. We did not find significant alignment of these axes with the lobule-level reference system, indicating that there is no nematic order for the cell body elongation (Figure 2—figure supplement 2). Regarding the correlation between cell shape and polarity axes, we only observed weak correlations between them (Figure 2—figure supplement 2).

3) The reference to a biaxial nematic order at the tissue scale (subsection “Spatial patterns of cell polarity reveal liquid-crystal order in the liver lobule”) is also unclear. This seems to suggest that one direction perpendicular to the flux orientation J is singled out, while this does not seem to be the case here.

In fact, a number of observations within our study suggest nematic order inside liver tissue that is not only uniaxial, but biaxial.

- As one example, the correlation between the second nematic axis a_2_ and the reference field J is different from the baseline, which one expects for pure uniaxial order.

- Moreover, the layered order of liver tissue characterized in Figure 3H singles out a second reference axis perpendicular to the flux orientation J, which, together with the flux direction J, spans the layers. This second reference axis is approximately parallel to the axis of the central and portal vein.

We acknowledge that biaxial order is not yet formally quantified in the current manuscript. We modified the text to clarify this point (Results subsection “Spatial patterns of cell polarity reveal liquid-crystal order in the liver lobule”). Additionally, in a separate study (Scholich et al., 2019), we extensively characterized biaxial order of hepatocyte cell polarity with major focus on mathematical aspects of the problem. This quantitative analysis confirms the biaxial nature of hepatocyte polarity. We referenced this study in the manuscript (Results subsection “Spatial patterns of cell polarity reveal liquid-crystal order in the liver lobule”).

4) Where is the evidence to show that the silencing of Integrin-β1 does not affect the sinusoidal cells? This is an important point to support the claim that the large-scale tissue organisation requires two-way communication between hepatocytes and endothelial cells.

It has previously been shown that the injection of siRNAs formulated into lipid nanoparticles (LNP) targets specifically hepatocytes in adult mice (Bogorad et al., 2014; Zeigerer et al., 2012). To corroborate this conclusion, we analyzed Integrin-β1 expression and sub-cellular localization by using confocal and super-resolution microscopy to resolve the plasma membrane of hepatocytes from that of sinusoidal endothelial cells (Figure 4—figure supplement 1). Integrin-β1 expression was ablated in both the basal and lateral plasma membrane of hepatocytes but not in the plasma membrane of the sinusoidal endothelial cells. The text was modified accordingly (Results subsection “Bidirectional communication between hepatocytes and sinusoids is necessary for the maintenance of tissue structure”).